# *CHEK2* Alterations in Pediatric Malignancy: A Single-Institution Experience

**DOI:** 10.3390/cancers15061649

**Published:** 2023-03-08

**Authors:** Eman Abdelghani, Kathleen M. Schieffer, Catherine E. Cottrell, Anthony Audino, Kristin Zajo, Nilay Shah

**Affiliations:** 1Division of Hematology/Oncology/Blood and Marrow Transplantation, Nationwide Children’s Hospital, Columbus, OH 43205, USA; 2The Steve and Cindy Rasmussen Institute for Genomic Medicine, Nationwide Children’s Hospital, Columbus, OH 43215, USA; 3Department of Pediatrics, The Ohio State University College of Medicine, Columbus, OH 43210, USA; 4Department of Pathology, The Ohio State University College of Medicine, Columbus, OH 43210, USA

**Keywords:** *CHEK2*, cancer, cancer predisposition, pediatric oncology, genetic counseling

## Abstract

**Simple Summary:**

Pediatric malignancies rarely occur, with 12,000–15,000 cases per year in the United States. Approximately 10% of pediatric cancers are thought to be secondary to germline alterations in cancer-predisposing genes based on numerous germline genomic analyses. *CHEK2* germline loss-of-function variants have been reported in certain pediatric cancer patient cohorts, including neuroblastomas, non-Hodgkin lymphomas, thyroid cancer, melanomas, sarcomas, and brain tumors. However, a paucity of data exists surrounding clinical phenotypes and outcomes. In this study, we present our single-institution experience regarding six children and adolescents with *CHEK2* germline alterations and cancer or cancer-predisposing conditions, including their clinical presentations and outcomes. We also review the current body of data regarding *CHEK2* germline alterations in the pediatric cancer population and future challenges with studies on these alterations.

**Abstract:**

Background: Approximately 10% of pediatric malignancies are secondary to germline alterations in cancer-predisposing genes. Checkpoint kinase 2 (*CHEK2*) germline loss-of-function variants have been reported in pediatric cancer patients, but clinical phenotypes and outcomes are poorly described. We present our single-institution experience of pediatric oncology patients with *CHEK2* germline alterations, including clinical presentations and outcomes. Methods: Pediatric oncology patients with *CHEK2* germline alterations were identified among those assessed by clinical or translational research at the Institute for Genomic Medicine at Nationwide Children’s Hospital. A chart review of disease course was conducted on identified patients. Results: We identified 6 patients with germline *CHEK2* variants from a cohort of 300 individuals, including 1 patient with concurrent presentation of Burkitt lymphoma and neuroblastoma, 3 patients with brain tumors, 1 patient with Ewing sarcoma, and 1 patient with myelodysplastic syndrome. Three patients had a family history of malignancies. Four patients were in remission; one was undergoing treatment; one patient had developed treatment-related meningiomas. We review prior data regarding *CHEK2* variants in this population, challenges associated with variant interpretation, and genetic counseling for individuals with *CHEK2* variants. Conclusions: *CHEK2* germline loss-of-function alterations occur in patients with a variety of pediatric tumors. Larger multicenter studies will improve our understanding of the incidence, phenotype, and molecular biology of *CHEK2* germline variants in pediatric cancers.

## 1. Introduction

Pediatric malignancies are uncommon in the United States, with 12,000–15,000 cases per year [1]. Historically, the genetic underpinnings of these cancers were undefined. However, through the use of high-throughput next-generation sequencing (NGS) in current clinical and translational research, numerous somatic genomic alterations, including single nucleotide variants, small insertions-deletions, and gene fusions, have been identified within many of these tumors [2,3,4,5]. Additional germline genomic analyses of children with cancer have shown that nearly 10% of patients harbor germline alterations in cancer-predisposing genes thought to be related to their disease [6,7,8,9,10,11]. Among these analyses, select genes have been shown to be more recurrently altered in the germline genomic profiles of pediatric cancer patients, though the clinical phenotypes and outcomes associated with most of these alterations have been poorly described. 

Checkpoint kinase 2 (*CHEK2*), located at chromosome 22q12.1, encodes a protein kinase that is activated by phosphorylation by the ATM protein in response to DNA damage. It serves multiple functions including TP53 stabilization, cell cycle arrest, genome maintenance, and apoptosis [12,13]. A multiorgan cancer susceptibility gene, *CHEK2*, is one of the most frequently identified genes harboring germline alterations in patients with various cancers [12,14,15], including in children, adolescents, and young adults [4,16]. However, clinical interpretation of these variants can be challenging, in part due to their high frequency in unaffected founder populations (e.g., c.1100delC and p.I157T with 0.02–1.5% and 1–5% frequency across selected European populations, respectively) with incomplete segregation and limited characterization of other variants particularly in under-represented populations [17]. Prior studies have predominately focused on the c.1100delC founder variant [12]. The current literature suggests that *CHEK2* variants may portend a mild-to-moderate risk for various malignancies, including breast, colorectal, and prostate cancer in adults, dependent on the specific alteration. Additionally, germline *CHEK2* variants were also shown to be associated with cancer-predisposing disease, such as myelodysplastic syndrome [18]. 

*CHEK2* germline loss-of-function alterations have been reported at low frequencies in high-throughput analyses of pediatric cancer cohorts, including neuroblastomas, non-Hodgkin lymphomas, thyroid cancer, melanomas, sarcomas, and brain tumors [2,19,20,21,22,23,24], but the associated clinical phenotypes and outcomes remain poorly described or understood. We present our single-institution experience with six children with *CHEK2* germline alterations and cancer or cancer-predisposing conditions, including their clinical presentations and outcomes.

## 2. Materials and Methods

### 2.1. Patient Enrollment

Patients with *CHEK2* germline alterations were identified from 246 individuals with solid tumors, hematologic malignancies, or myelodysplasias who underwent sequencing as part of an institutional translational protocol from 1 December 2017 to 30 June 2021. This IRB-approved protocol (IRB17-00206) involved comprehensive molecular profiling including exome sequencing of paired disease-involved specimens and germline tissue RNA sequencing of disease-involved specimens; and, when appropriate, DNA methylation array-based classification. This protocol allowed for the identification of underlying genomic alterations and fusions that may be drivers of and/or predisposing to disease. Additionally, 54 individuals with clinical paired tumor/normal exome sequencing were included in this cohort, resulting in a total cohort of 300 individuals. Chart review was conducted through the electronic medical record of those with identified *CHEK2* variants to review disease course. Patients and their legal guardians provided informed consent (and assent, when appropriate) at time of enrollment on the translational protocol or consent for clinical testing. 

### 2.2. Enhanced Exome Sequencing 

Enhanced exome sequencing was performed on DNA extracted from a comparator germline sample (peripheral blood, saliva, or buccal swabs) and disease-involved tissue (snap frozen tumor tissue, formalin-fixed paraffin-embedded (FFPE) tumor tissue, or bone marrow). An estimate of tumor cellularity was obtained by histologic assessment of a hematoxylin and eosin stained section of tissue. Libraries were prepared using 100–500 ng of input DNA beginning with enzymatic fragmentation, followed by end repair, 5′ phosphorylation, A-tailing, and sequencing adapter ligation using NEBNext Ultra II FS reagents (New England Biolabs, Ipswich, MA, USA). Target enrichment by hybrid capture was performed with IDT xGen Exome Research Panel v1.0 (Individuals 3–5) or v2.0 (Individuals 1, 2, 6), enhanced with xGenCNV Backbone and Cancer-Enriched Panels-Tech Access (Integrated DNA Technologies, Coralville, IA, USA). Paired-end 151 bp reads were generated on an Illumina HiSeq 4000 (Individuals 1 and 2) or NovaSeq (Individuals 3–6). Secondary analysis was performed using Churchill, a comprehensive workflow for analysis of raw reads from genome alignment through to germline and somatic variant identification [21]. Reads were aligned to the human genome reference sequence (build GRCh37 (Individuals 1–5) or GRCh38 (Individual 6) using BWA (v0.7.15). Sequence alignments were refined according to community-accepted guidelines for best practices [25]. Duplicate sequence reads were removed using samblaster-v.0.1.22, and local realignment was performed on the aligned sequence data using the Genome Analysis Toolkit (v3.7-0). Churchill’s own deterministic implementation of base quality score recalibration was used. Germline variants were called using GATK’s HaplotypeCaller. Average sequencing coverage depth for the six samples was as noted: germline (Individual 1: 246×, Individual 2: 336×, Individual 3: 247×, Individual 4: 206×, Individual 5: 155×, and Individual 6: 191×) and somatic (Individual 1: 246× (lymphoma) and 108× (neuroblastoma), Individual 2: 374×, Individual 3: 255×, Individual 4: 299×, Individual 5:219×, and Individual 6:229×) samples. Germline variants in cancer-associated genes were identified [6]. Somatic single-nucleotide variation (SNV) and indel detection were performed using MuTect-2 [26]. Copy number alterations (CNAs) were assessed using VarScan2 and GATK-CNV [27].

## 3. Results

We identified six individuals with solid tumors or hematologic disease with germline *CHEK2* alterations at Nationwide Children’s Hospital between 1 December 2017 and 1 November 2022. The clinical history and genomic findings from the six individuals are described below and are summarized in Table 1.

Individual 1: A 6-year-old boy who presented with neck swelling was diagnosed with stage III high-risk Burkitt lymphoma with bone and bone marrow (immunophenotype only) involvement. Tumor cytogenetics revealed a 46,XY,t(8;22)(q24.2;q11.2)[cp19]/46,XY[1] karyotype, consistent with a *MYC* rearrangement. Fluorescence in situ hybridization (FISH) using a *MYC* breakapart probe demonstrated split signals in 89.5% of cells examined (nuc ish (MYCx2)(5′MYC sep 3′MYCx1)[179/200],(BCL6,IGH,BCL2)x2[200]). He received chemotherapy as per Children’s Oncology Group (COG) protocol ANHL1131 [28] Group B, using chemotherapy and the anti-CD20 antibody rituximab. He underwent prephase chemotherapy (one cycle), induction chemotherapy (two cycles), and the first of two cycles of consolidation chemotherapy. Disease response evaluation by PET-CT demonstrated response of most sites but persistent avidity in a midabdominal para-aortic mass (standardized uptake value 2.4; Deauville score: 4 [29]). Excision and pathologic evaluation of the mass identified a neuroblastic tumor consistent with poorly differentiating neuroblastoma with post-treatment effect, with no evidence of Burkitt’s lymphoma. Clinical genomic evaluation of the tumor showed no alterations in the *MYCN* or *ALK* genes. Tissue chromosomal microarray analysis identified whole chromosomal gains in the absence of segmental gains or losses, consistent with a favorable triploid tumor genome. Complete staging, including imaging and marrow biopsy, demonstrated no metastatic disease, and the surgery had rendered the patient with no evidence of disease. As such, no further neuroblastoma treatment was indicated. The patient completed consolidation chemotherapy, and end-of-therapy evaluation confirmed the patient to be in complete remission. Genomic evaluation revealed a germline *CHEK2* (NM_007194.4) c.1100delC (p.Thr367fs) heterozygous frameshift alteration. Familial testing was not performed, and no family history of breast cancer was noted. Additionally, in the lymphoma sample, an *IGLL5::PVT1* fusion was identified by RNA sequencing, consistent with the t(8;22)(q24.2;q11.2) rearrangement reported by cytogenetics. Somatic variants in lymphoma-associated genes were also identified: *DDX3X* p.Gln27*, *IGLL5* p.Trp7Ter, *MYC* p.Val6Ile, *MYC* p.Val7Met, *MYC* p.Gln49Arg, and *MYC* p.Thr73Ser. This pattern of somatic findings is consistent with the tumor type under study. He remains in remission 18 months off therapy.

Individual 2: An 11-year-old boy with headaches and vomiting was diagnosed with a pilocytic astrocytoma of the posterior fossa, WHO Grade 1. He underwent near-total resection and ventriculostomy without adjuvant therapy. He suffered sequelae of diplopia, imbalance, cerebral salt wasting, and precocious puberty. Fifteen months later, he had an asymptomatic local recurrence of the astrocytoma identified on surveillance MRI with associated hydrocephalus. A ventriculoperitoneal shunt was placed to decompress the hydrocephalus, and the tumor was then again resected. Pathology again demonstrated a WHO Grade I pilocytic astrocytoma. Three months later, he was found to have leptomeningeal and spinal cord metastases. He received craniospinal irradiation, with 36 Gy to all sites and boosts of 14.4 Gy to the original tumor bed and 5.4 Gy to the lower spine. Post-treatment evaluation demonstrated control of the astrocytoma but other stable lesions considered to be radiation-associated meningiomas that could be observed. Five years later, surveillance imaging revealed a new right cerebellar tentorial lesion considered most likely a meningioma; this lesion and the prior meningiomas were monitored over the next two years with slow growth but no symptoms. After 2 years of observation, the right tentorial mass was resected and identified as an atypical meningioma, WHO Grade 2. At this point, the patient was consented for our institutional translational protocol. A *CHEK2* c.1100delC (p.Thr367fs) heterozygous germline frameshift variant was identified. This variant was maternally inherited and the proband’s mother had a history of breast cancer. The meningioma was also sequenced and found to have an *NF2* p.Glu34fs frameshift variant (predicted to encode a premature stop of translation) and loss of chromosome 22q. Notably, *CHEK2* and *NF2* map to chromosome 22, with *CHEK2* at 22q12.1 and *NF2* at 22q12.2. Inactivating alterations in *NF2* with subsequent chromosome 22 loss is frequently described among meningiomas. Within this tumor, the deletion of the long arm of chromosome 22 effectively resulted in reduction to homozygosity of the *NF2* frameshift variant but loss of the *CHEK2* mutant allele, resulting in enhanced representation of the wild-type allele in the tumor. This individual is currently enrolled on a clinical trial for treatment of local progression of the meningiomas.

Individual 3: A 7-year-old boy who presented with progressively worsening headache was found on MRI to have a mass in the left lateral ventricle measuring 5.4 × 3.4 × 5.6 cm, centered around the left foramen of Monro and avidly enhancing with no clear restricted diffusion or enhancement. He underwent gross total resection; the tumor pathology was consistent with subependymal giant cell astrocytoma, WHO Grade I. Germline *TSC1*/*TSC2* gene sequencing was negative. Therefore, he was consented on our institutional translational protocol, which revealed a germline heterozygous *CHEK2* c.444+1G>A splice site alteration. Familial testing was declined; however, there is a family history of early breast cancer in the maternal grandmother, prompting proactive breast imaging in the proband’s mother, who is currently asymptomatic. Somatic tumor analysis additionally demonstrated a *TSC1* c.1525C>T (p.Arg509Ter) nonsense variant and LOH of 9q, resulting in a reduction to homozygosity of the *TSC1* p.Arg509Ter variant. This constellation of findings is consistent with subependymal giant cell astrocytoma. This individual remains in remission 23 months after resection.

Individual 4: A 6-year-old boy who presented with progressively worsening headaches and intractable emesis was found on MRI to have a midline posterior fossa mass with obstructive hydrocephalus. He underwent gross total resection and was diagnosed with a medulloblastoma (WHO Grade IV, classic histology, non-WNT/non-SHH subtype). Immunohistochemistry was negative for YAP1, GAB1, nuclear beta-catenin, and TP53. FISH probes hybridized to *MYC* and *MYCN* demonstrated no amplification. He underwent proton craniospinal irradiation and then maintenance chemotherapy per ACNS0331 [30]. The patient was consented for our institutional translational protocol, which revealed a germline heterozygous *CHEK2* c.470T>C (p.Ile157Thr) missense variant. This missense variant was shown to result in impaired oligomerization and autophosphorylation of the CHEK2 protein [31]; however, functional data are conflicting [32]. Follow-up testing revealed this variant to be maternally inherited. While the proband’s mother is currently asymptomatic, there is a history of breast and endometrial cancer in her extended family. Somatic analysis of the disease-involved sample was consistent with non-WNT/non-SHH medulloblastoma, including *ZMYM3* p.Ala217fs and *SETD2* p.Glu1582Lys variants, in addition to copy number gain of chromosome 7, loss of chromosome 8 and 13, and isochromosome 17q. This individual remains in remission 12 months after therapy.

Individual 5: A 15-year-old male who presented with pancytopenia was diagnosed with severe idiopathic aplastic anemia. He responded well to immunosuppressive therapy with horse antithymocyte globulin (ATG) and maintenance cyclosporine. Twenty-eight months after the initiation of immunosuppression, surveillance bone marrow evaluation demonstrated dysplastic features diagnostic of myelodysplastic syndrome (MDS). The patient was consented on our institutional translational protocol, which revealed a germline heterozygous *CHEK2* c.283C>T (p.Arg95Ter) nonsense variant. Familial testing was not performed, and the pedigree did not reveal a history of cancer. A somatic variant in the *PIGA* gene (c.981+1G>A) was consistent with paroxysmal nocturnal hemoglobinuria. He underwent allogenic bone marrow transplant (BMT) that was complicated by primary graft failure and subsequent autologous reconstitution with 100% recipient cells. Despite this, he remains in remission 28 months after BMT without evidence of disease or cytopenias.

Individual 6: A 16-year-old male presented with right thigh and knee pain and limping. A soft mass was palpable on the anteromedial thigh. An MRI was performed, which showed an aggressive lesion of the right femoral diaphysis extending into the distal metaphysis and a chest CT showed multiple lung nodules. A biopsy of the femoral lesion was consistent with Ewing sarcoma, with confirmation by reverse-transcription polymerase chain reaction (RT-PCR) of tumor RNA demonstrating *EWSR1*::*FLI1* gene fusion. He received treatment as per COG protocol AEWS1031 [33], in addition to limb salvage resection for local control, with a complete response. He presented with a localized recurrence of soft tissue disease at the original primary tumor site 14 months after initial therapy. He was treated with vincristine, irinotecan, and temozolomide as per COG protocol ARST08P1 and radiotherapy [34]. The patient consented for clinical paired tumor/normal exome sequencing, which revealed a germline heterozygous *CHEK2* c.1100delC (p.Thr367fs) frameshift variant. A review of the pedigree revealed a history of early breast cancer and colon cancer in multiple siblings of the maternal grandmother and first cousins once removed. The mother’s maternal cousin had genetic testing, which revealed a *PALB2* variant, although the specific variant information was unavailable. Cascade testing of the *CHEK2* variant was recommended during the genetic counseling visit. Paired tumor/normal exome sequencing also identified the presence of an *ERF* p.Met76IlefsTer5 somatic variant and multiple copy number alterations, consistent with a diagnosis of Ewing sarcoma.

In summary, the germline *CHEK2* alterations presented in individuals with a diversity of tumor histologies, including one individual with concurrent Burkitt lymphoma and neuroblastoma. Interestingly, all patients in our cohort harboring germline *CHEK2* variants are male. 

## 4. Discussion

Herein, we report on six pediatric patients diagnosed with malignancy or hematologic disease and subsequently observed to harbor germline *CHEK2* alterations. Amid a total cohort size of 300, these data highlight the spectrum and frequency of germline *CHEK2* variants detected in our pediatric oncology population. Our data demonstrate that germline *CHEK2* alterations are identified in patients with a variety of neoplastic or clonal somatic processes with some unique features.

*CHEK2* encodes a kinase that functions as a tumor suppressor, which is activated in cells under circumstances of DNA damage and other cellular damage [12]. *CHEK2* is not necessary for viability in animal models but increases tumor incidence [35], suggesting that even partial loss of expression may impair cellular response to DNA damage. This is supported by studies showing variable rates of somatic mutations or loss of heterozygosity of the remaining *CHEK2* allele in adult breast cancer patients [36,37]. As such, *CHEK2* germline alterations may contribute to oncogenesis through the inability to activate DNA damage repair pathways and/or the accumulation of additional genetic variants, leading to tumor formation and metastasis [38]. Additional work is needed to determine if there are patterns of alterations that contribute to specific malignancies, as that will help manage any appropriate screening guidelines for these patients.

Five patients in this case series were diagnosed with a solid tumor, including three with a brain tumor, one with neuroblastoma, and one with Ewing sarcoma. Germline *CHEK2* variants are considered rare in pediatric cohorts [39], but childhood cancer patients have historically not been assessed for such variants, leading to a potential underestimation of the true prevalence. Regardless, no prior studies have specifically reported germline *CHEK2* alterations in subependymal giant cell tumors or medulloblastomas, although *CHEK2* germline alterations have been identified in patients with glioblastoma [20,40] and other pediatric brain tumors [41,42,43]. Of note, *CHEK2* variants of uncertain significance (VUS) were previously described [43], but their effect, if any, is unclear in relation to pathogenesis. Nominal enrichment of *CHEK2* variants have been described among individuals with Ewing sarcoma [44,45]. The role of *CHEK2* alterations in hematological malignancies are under study. Janiszewska et al. [46] found that patients with MDS had a higher incidence of germline *CHEK2* variants than healthy controls, and these patients also had a poorer prognosis than patients with MDS without *CHEK2* alterations. Sharifi et al. [47] showed that promoter methylation and decreased expression of *CHEK2* in patients with MDS was associated with worse clinicopathological features. The p.Arg95Ter *CHEK2* variant, resulting in a missense alteration, reported in Individual 5, was previously reported in endometrial carcinoma and choriocarcinoma [48], but never previously in MDS. Additionally, Stubbins et al. recently reported a potential association with myeloid malignancies and MDS and *CHEK2* germline variants [49].

Notably, Individual 1 had a complex clinical presentation, with concurrent diagnosis of neuroblastoma and Burkitt lymphoma. To the best of our knowledge, there are no previous reports of a patient with lymphoma and neuroblastoma concurrently identified. The timeline of oncogenesis of each cancer, i.e., if metachronous or synchronous tumor development occurred, remains unclear. The histology and chromosomal findings of the neuroblastoma suggest a low-grade tumor, which may have been incidentally present for some time, as these tumors relatively commonly develop during infancy and may persist for years. Burkitt lymphomas, in contrast, are aggressive, rapidly proliferating tumors; as such, the lymphoma more likely developed in temporal proximity to clinical presentation. Nonetheless, *CHEK2* germline alterations have been associated with both diseases. Pugh et al. [2] evaluated 240 neuroblastoma tumors and matched blood samples by genome sequencing, identifying 3 (1.2%) individuals with *CHEK2* germline variants (p.Arg145Trp, p.Arg181His, and p.Arg180His). Laorsa et al. [4] performed exome sequencing on 17 patients with aggressive neuroblastoma tumors and reported 3 additional germline alterations (p.Leu174Phe, c.793-1G>A, p.Asp438Tyr). Studies have reported pathogenic germline *CHEK2* loss-of-function alterations in non-Hodgkin lymphomas, including Burkitt lymphomas [50,51]. The c.1100delC alteration found in Individual 1 was previously reported in other non-Hodgkin lymphomas [19], but this is the first report of this germline variant in a patient with Burkitt lymphoma or neuroblastoma.

The literature [45,46] reviewing cancer risk in children with germline *CHEK2* variants is limited. *CHEK2* alterations have been reported in a breadth of tumor types, including those not encountered in this case series, including papillary thyroid carcinomas [52] and Wilms tumors [11,53]. However, historically, the majority of patients with pediatric tumors, generally and including at our institution, have not undergone testing for cancer-predisposing genomic variants. Furthermore, tumor genomic profiling assays are variable in the breadth of included gene content, usually with a focus on somatic variants, with no independent assessment of a germline comparator sample. As such, there exists an ascertainment bias with respect to prior knowledge of cancer predisposition, etiology of the variant (germline vs somatic), and the population under study. As genomic evaluations are more extensively performed, the full spectrum of malignancies associated with *CHEK2* alterations will be better defined.

Presently, the contribution of the germline *CHEK2* variant to predisposition to the cancers described in this case series is unclear. We identified germline *CHEK2* variants in 2% of our pediatric/young adult cohort, with no clear discernment toward a particular tumor type. Biallelic alteration of *CHEK2* was not identified within this cohort, although biallelic germline or somatic alterations are uncommonly found in adults with *CHEK2*-associated cancers [54,55,56]. Nonetheless, the variants identified in our study may represent secondary findings amid individuals in a pediatric cancer cohort. Notably, the genomic profiling for Individual 2, who presented with an atypical meningioma with a chromosome 22 loss, resulted in loss of the mutant *CHEK2* allele harboring the c.1100del variant in the tumor and with retention of the wild-type *CHEK2* allele in a single-copy state. Co-deletion of *CHEK2* and *NF2* (both on chromosome 22) was reported to contribute to meningioma pathogenesis and genomic instability [57]. As such, increased use of tumor genomic profiling assays in pediatric cancer patients may help elucidate the relevance of *CHEK2* variants in these cancers, alone and in combination with other genetic events.

Due to many factors, including high prevalence of founder variants in healthy individuals and incomplete penetrance within families, clinical interpretation of germline *CHEK2* variants is challenging. Furthermore, ascertainment bias across both tumor type and age of individuals under study, and limited germline screening to-date in children may contribute to the lack of information regarding cancer risk in the pediatric setting. Classification of *CHEK2* variants using standards set forth by the American College of Medical Genetics and Genomics (ACMG) and Association for Molecular Pathology (AMP) proves challenging, with many laboratories reporting conflicting interpretations of pathogenicity [58,59]. Within the ClinVar database (accessed 25 November 2022), 65 coding *CHEK2* variants are reported with conflicting interpretations of pathogenicity, with 24 (37%) in conflict as pathogenic/likely pathogenic or variant of uncertain significance (VUS) and 37 (57%) in conflict as VUS or benign/likely benign [60]. Of the total coding variants, missense variants are most frequently reported with conflicting interpretations of pathogenicity (53/65, 82%) or as a VUS (1284/1299, 99%). While there have been some functional studies to assess the pathogenicity of *CHEK2* missense variants, larger studies are necessary to permit application of specific ACMG/AMP criteria (i.e., PS3) [32,59,61,62,63]. In the setting of paired tumor/normal sequencing inclusive of broad cancer-associated gene content and allowing for comprehensive analysis of the *CHEK2* gene, variant classification may improve, further delineating which *CHEK2* variants may impact cancer risk [64,65,66].

Although genetic testing for conditions traditionally considered to represent “adult-onset” cancer predisposition has historically been discouraged in the pediatric cancer population, this paradigm is becoming increasingly challenged by the increased uptake of broad genetic testing such as exome or genome sequencing in pediatric patients, the ability to infer germline results from somatic testing, and the fact that therapies targeting specific mutations or pathways are becoming increasingly available for pediatric patients. Current guidelines for screening of adults with germline *CHEK2* alterations include the consideration of annual breast MRI beginning at age 35–40, annual mammograms beginning at age 40 or 5–10 years before the earliest breast cancer diagnosis in the family, and colonoscopy every 5 years beginning at age 40 or 5–10 years before the earliest colon cancer diagnosis in the family [67,68]. However, given that we are testing a pediatric population, many of the patients’ first-degree relatives are relatively young at the time of diagnosis and may not have developed a canonical *CHEK2*-related cancer at the time of testing. This, in conjunction with the variable penetrance of *CHEK2* variants, suggests that for many families, the presence of a *CHEK2* variant may be more informative in terms of recommended cancer surveillance than family history alone. Genetic counseling has been important in our identified patient population to better assess (1) how families interpret the ambiguity surrounding the contribution of the *CHEK2* alteration to their child’s current cancer diagnosis, (2) how this information alters their perceived cancer risks for their child and other family members, and (3) to help guide decision making regarding follow-up testing and screening. For example, parents often express a desire to test siblings of the affected individual; however, such testing is not otherwise recommended in healthy minors, as there are currently no medical management guidelines for individuals with pathogenic *CHEK2* variants under 18 years of age [67,68]. For some families, knowledge of a *CHEK2* alteration in an otherwise healthy child without any recommended cancer screening in childhood may lead to increased distress. Discussing the benefits and limitations of testing form a psychological, financial, and ethical perspective is a key aspect of pre- and post-test counseling for pediatric cancer patients and their families. These conversations and recommendations regarding testing and cancer screening for *CHEK2* alterations will likely continue to evolve over time as we better understand their contribution to adult and pediatric cancers.

## 5. Conclusions

*CHEK2* germline pathogenic alterations can be seen in a variety of pediatric tumors. Due to historic ascertainment bias in the population under study for cancer predisposition, as well as limitations in tumor genomic profiling assay content and design, the frequency and clinical meaning of *CHEK2* variation have not been fully defined. Broader studies will improve our understanding of the true incidence of these variants and of the biologic contribution of *CHEK2* germline and somatic alterations in pediatric cancers. Secondary findings detected in the setting of tumor genomic profiling, particularly in the setting of paired tumor/normal analyses, are gaining increasing recognition. Over time, as genomic testing evolves and increased long-term follow-up of individuals in pediatric and adult populations with *CHEK2* variants occurs, greater knowledge of surveillance and cancer risk may become known. This will also inform guidance as to the testing of family members of affected patients and their potential disease risks.

## Figures and Tables

**Table 1 cancers-15-01649-t001:** Clinical and genomic findings of patients with CHEK2 alterations.

Individual	Age at Diagnosis (yr)	Sex	Diagnosis	Family History	Germline *CHEK2* (NM_007194.4) Alteration	Somatic Variants	Somatic Copy Number Alterations (CNAs)	Clinical Outcome	Time of Last Follow Up
1	5	M	Burkitt’s lymphoma and concurrent neuroblastic tumor	+ Family history of bladder, pancreatic and bone cancer	c.1100delC (p.Thr367fs)	Lymphoma: *IGLL5*::*PVT1* fusion, *DDX3X* p.Gln27Ter, *IGLL5* p.Trp7Ter, *MYC* p.Val6Ile, *MYC* p.Val7Met, *MYC* p.Gln49Arg, *MYC* p.Thr73Ser Neuroblastoma: no somatic variants identified	Lymphoma: no somatic CNAs identified Neuroblastoma: Gains: 2, 4, 6, 7, 8, 9, 10, 12, 13, 15, 17, 18, 20, 21	Lymphoma was treated with chemotherapy. Neuroblastic tumor was resected surgically only. Patient is in remission	30 months off therapy
2	11	M	Meningioma	No family history of cancer	c.1100delC (p.Thr367fs)	*NF2* p.Glu34fs	Gains: segmental 2q, telomeric 7p Losses: 1p, 2p, segmental 2q, interstitial 4q, 6q, telomeric 9q, 18q, 22q	Astrocytoma initially resected; recurred locally and with spinal metastasis. Received CSI, developed secondary meningiomas	Currently continuing treatment for meningioma
3	7	M	Subependymal giant cell tumor	No family history of cancer	c.444+1G>A	*TSC1* p.Arg509Ter	cnLOH: 9q (including *TSC1*)	Tumor treated with gross total resection	34 months from tumor resection
4	6	M	Medulloblastoma	+ Family history of soft tissue cancer	c.470T>C (p.Ile157Thr)	*ZMYM3* p.Ala217fs; *SETD2* p.Glu1582Lys	Gains: 7 Losses: 8, 11, 13, 14 Isochromosome 17q	Tumor treated with gross total resection, proton CSI and chemotherapy	20 months off therapy
5	15	M	Aplastic anemia/MDS	No family history of cancer	c.283C>T (p.Arg95Ter)	*PIGA* c.981+1G>A	No somatic CNAs identified	MDS treated with allogenic bone marrow transplant with graft failure, remains in remission	31 months after BMT
6	16	M	Ewing sarcoma	+ Family history of breast and colon cancer	c.1100delC (p.Thr367fs)	*ERF* p.Met76Ilefs	Gains: 1q, 8, 14, 18, 21 Losses: 7p, *CDKN2A/B* biallelic loss	Tumor treated with chemotherapy and limb salvage resection for local control. He presented with relapsed disease 14 months post-treament	Undergoing salvage chemotherapy and radiation therapy for relapsed disease

MDS: myelodysplastic syndrome, cnLOH: copy-neutral loss of heterozygosity, CSI: craniospinal irradiation, BMT: bone marrow transplant.

## Data Availability

Sequencing data in relation to this study were deposited to dbGAP, accession number phs001820.v1.p1 (https://www.ncbi.nlm.nih.gov/projects/gap/cgi-bin/study.cgi?study_id=phs001820.v1.p1).

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
