# Peer review of "CHEK2 Alterations in Pediatric Malignancy: A Single-Institution Experience"

_cancers, 2023, doi:10.3390/cancers15061649_

Round 1

Reviewer 1 Report (Previous Reviewer 2)

I finished reviewing the revised version of the manuscript and I also looked up the author's reply.   The manuscript is fully revised and the authors made lots of changes according to my comments and suggestions. As the authors significantly improved the manuscript and the academic editor suggested a resubmission after important changes and improvements, I believe that the revised version could be suitable for publication.

Reviewer 2 Report (New Reviewer)

I would recommend to accept the revised manuscript

This manuscript is a resubmission of an earlier submission. The following is a list of the peer review reports and author responses from that submission.

Round 1

Reviewer 1 Report

CHEK2 is a commonly mutated gene in cancer, identifying it in the germline of a child with cancer does not in itself indicate that it is driving tumorigenesis, in fact, it is considered an adult onset cancer gene causing Hereditary Breast and Ovarian cancer syndrome. There are some CHEK2 variants that are at high frequency in the general population (1.5%), and considered low penetrance for adult breast cancers. These nuances need to be presented to put the identification of CHEK2 variants in context. Additionally, because genomic sequencing was done, and other potential drivers were identified, it is entirely possible that the CHEK2 variants had no relevance to tumorigenesis whatsoever, especially since no second hits were found.

Table 1: the c.1100delC variant is found in high frequency in the general population, is it enriched in your pediatric cohort?

Genomic sequencing: The usual mechanism of action of a tumour suppressor gene is Knudson’s two-hit hypothesis. Neither second mutations, nor LOH, has been identified in the tumours, therefore there is no evidence that it is contributing to cancer-risk.

Line 234: are these adult or pediatric patients? And please provide numbers and p-values, that would be helpful to the reader to gauge the significance, as currently MDS is not screened for in adults with CHEK2 pathogenic variants.

244: this is a stretch, I don’t think that increased tumour incidence in animal models (and this should be explained in some detail so the reader can judge for themselves) suggest that impairs cellular response to DNA damage.  This statement needs to have evidence associated to it, otherwise it’s speculation, and out of scope for a paper cataloging how many pediatric patients with CHEK2 have cancer at their institution.

Line 251; It may be notable, but is it statistically significant given the numbers? This could be easily calculated, and should be if they want to discuss a sex-bias. All of the HBOC genes increase the risk for male breast cancer, because men still have breast tissue, though less than women, hence less risk. It does not mean a ‘potentiated  effect’ on men, although I’m not exactly clear what a potentiated risk is.

Line 256: this can be easily looked up, and to my knowledge men doesn’t have an increased early life risk of cancer as compared to females, no need to guess here.

Line 261; in fact, there are clear screening guidelines from the NCCN on how to screen patients with CHEK2 pathogenic variants.

Line 267: the authors need to look up the population frequency of CHEK2 mutations to put this into context. The c.1100delC variant itself has a carrier frequency of up to 1.5% in North-West Europeans.

Reviewer 2 Report

Dear Authors,

Thank you for submitting the paper "CHEK2 mutations in pediatric malignancy: a single institution experience" to Cancers.

I have read your article carefully and, after an accurate review, I must reject it as ineligible for publication.

Surely, the topic is very interesting and can be important for the molecular characterization of pediatric tumors but there are many inaccuracies in the article especially in the materials and methods section and in the correlations between the variants and the clinical data. Furthermore, no significant data or functional experiments are reported to support the pathogenicity of the identified variants that could be considered causative.

In conclusion, I believe the article is ineligible for publication in Cancers

Major Revision

In the conclusions, the authors state that "the loss of germline function of CHEK2 mutations occur in patients with a variety of pediatric cancers." The variants identified in the CHEK2 gene in the manuscript are not all loss of function, they report a missense variant in patient 4.

The authors cannot make this kind of statement or conclusion which is also reported in the introduction

The materials and methods are inadequate and are described superficially. The experimental protocols that were used for the different analyzes conducted, which platforms were used, and which criteria were applied for the analysis (e.g. germline coverage, somatic coverage, resolution to identify CNA, exome capture, alignment, variant calling).

Table 1. Legend  is missing

The clinical description of the cases is quite detailed while important deficiencies in molecular data are highlighted. The genotype of the variants is reported only in patients 1 and 2, in the other 3 cases, it is not known whether the identified variant is in heterozygous or homozygous state. For the identified variants, no information regarding the population frequency databases (gnomAD, ExAC, dbSNP) is reported, while for the somatic variants it is not indicated whether these are present or described in other tumors.

It is not reported or not clear whether the variants are de novo or inherited from a parent, in the final part of the results the authors indicate that the patient 4 variant was inherited from the mother. Is the mother affected or healthy? The variant identified in patient 4 is a missense variant, a type of variant for which it is difficult to assign pathogenetic significance in the absence of functional studies. Authors should consider the clinical history of the mother of patient 4 in order to better interpret the identified variant.

In the discussion “Within our cohort, despite the variations in specific diagnoses, some patterns can be defined among the patients” Please specify which patterns.

“It is likely that, as more patients are identified and corresponding phenotypes delineated, these VUSs could be reclassified ad pathogenic”

This is not strictly correct, it is certainly true that the number of patients with delineated phenotypes who share variants in the same gene can help to define a variant as pathogenetic but it is enough. The authors with this statement lead us to think that only increasing the number of cases, could help to classify a VUS as pathogenetic. To reclassify a variant from VUS to pathogenetic, it is necessary to associate functional studies that can specifically describe the effect of the variant on the protein expression or function.

“It is notable that all five patients identified in our cohort are male. CHEK2 germline mutations……”

CHEK2 gene maps to chromosome 22, an autosome, so there are no differences in gene dosage between men and women as happens for genes located on chromosome X.

5 patients are a very small cohort to speculate that there may be an increased risk in males than females. Furthermore, as the authors indicate, the CHEK2 gene is a gene associated with breast cancer in both males and females without however showing greater enrichment in males than females or vice versa. There is no scientific evidence to state that the CHEK2 variants in males have an increased susceptibility to cancer compared to females.